# The Moderating Effect of Family Business Ownership on the Relationship between Short-Selling Mechanism and Firm Value for Listed Companies in China

**Wenzhen Mai \* and Nik Intan Norhan Binti Abdul Hamid**

Azman Hashim International Business School, Universiti Teknologi Malaysia, Johor Bahru 81310, Malaysia;
m-norhan@utm.my
\* Correspondence: wenzhen@graduate.utm.my

**Abstract:** This study demonstrates an investigation of the external corporate governance effect of short selling mechanisms on firm value in the Chinese context. The effect of family businesses is also examined as a moderator of the relationship between short-selling and firm value. Using panel data analysis of Chinese listed companies, this paper tests a total sample of 22,468 firm-year observations from the Shanghai and Shenzhen Stock Exchange from 2009 to 2019 by applying the PSM-DID method in order to mitigate self-selection and endogenous problems caused by the uniqueness of Chinese short selling mechanisms. The findings suggest that both deregulation and the propensity of short selling can improve the firm value. Our findings also established that family ownership weakens firm value with the availability of short-selling, which indicates that family businesses have long orientations and conduct better corporate governance practices than non-family business, as short-selling shows a weaker external governance effect on firm value creation by family businesses in China. A robust test of alternative measurements is conducted and validated. This study provides significant insights for policymakers to consider in order to further relax short-selling constraints, which can act as effective external governance for better firm value creation, especially for non-family businesses in developing countries.

**Keywords:** short-selling; family ownership; firm value

## 1. Introduction

The significance of short-selling has been documented in the extensive and varied literature that investigates the effectiveness of short-sellers' supervisory and monitoring role on firm performance and value creation. Short sellers, as compared to ordinary individual investors, are more experienced traders with advanced information processing skills to uncover and profit from negative news of companies that otherwise might remain hidden (Massa et al. 2015). Scholars have previously focused on developed countries, but more research about short selling has been conducted in emerging markets such as China (Cai and Guo 2018; Chen et al. 2016; Chen et al. 2020a; Jiang and Chen 2019; Mai and Hamid 2021a; Meng et al. 2017; Zou et al. 2021). China provides a critical point of discussion in the capital market and business management because of its unique cultural and social values and a strong sense of hierarchical structure for family networks and businesses.

The Chinese financial regulatory board has loosened its stock of short selling restrictions since March 2010. This transformative reform to the capital market has brought about a breakthrough in China's A-share market, ending a history of the unilateral trading market for over 20 years. The loosening of short selling rules is encouraged through the gradual expansion of underlying stocks. For the first phase of deregulation implementation, 90 companies were selected in the list of shortable stocks by the Shanghai and Shenzhen Stock Exchanges. By December 2019, the implementation of short-selling business in both stock markets has grown exponentially, and qualified investors could short 1600 companies with

shorting positions exceeding 290 billion yuan in 2019, according to reported short-selling data from Shanghai and Shenzhen Stock Exchange official websites. The securities lending business has rapidly developed in China, so it is crucial to understand the implications of short selling activities on the firm's financial and management performance.

With the rapidly increasing trend in research on short selling, the existing literature has shown that short selling has significant deterrent and monitoring implications for financial management by the function of post-factum price discovery, which limits abusive, ex-ante management opportunistic misbehaviors (Hughes-Morgan and Ferrier 2017; Li 2018; Lu 2018; Mai and Hamid 2020). Brockman et al. (2020) and Liu et al. (2019) confirmed that managers of companies would increase the firm operating performance due to the threat of short selling to avoid being accused of giving an excessive compensation package. Scholars also demonstrated that the short-selling mechanism is also a motivator for corporate innovation, referring to its quality and quantity by enhancing innovation transparency among firms and optimizing their top managers' compensation efficiency (He and Tian 2018; Li et al. 2019). While the above literature provides insights for the determinants of superior management affected by short selling, the influence of short selling control relaxation on firm value creation by reducing the company's agency problems is still unknown. Hence, this research conducted an empirical examination of the relationship of short selling deregulation and firm value of listed companies.

Moreover, this study considers the uniqueness of the context of Chinese business. Like other Asian countries, the characteristics of the Chinese private business's ownership identity are mostly family-owned, which largely mitigate the problems incurred by state-owned companies (Yang et al. 2020). Based on previous research in developed countries, businesses with high family ownership tend to mitigate Type I agency problems with appropriate management involvement by family members (Miroshnychenko et al. 2020; Tseng 2020). In comparison, family businesses imply a different dimension of conflicts of interest between the family owner and minor shareholders, as family owners have the incentive to pursue family interests at the expense of minority shareholders (Pittino et al. 2020; Swanpitak et al. 2020b), which is known as a Type II agency problem. Gao et al. (2020) further analyzed that the unfavorable environment for institutions and weak protection in rules and law for minor shareholders' protection induces family owners to be involved in activities benefiting their private interests. For understanding family ownership in the Chinese business context, this study attempts to investigate how family ownership can moderate the relationship between short selling and firm value for listed family companies in China.

The main contributions of this paper are presented as follows. First, from the deterrence perspectives, short selling can prevent management misbehaviors and mitigate agency problem, although the effectiveness of short selling on firm creation remains unknown. This research further expands the existing literature to directly examine how the deregulation of short selling in China affects firm value creation by assessing the Tobin's Q and ROA of listed companies as measurements. Second, as most private businesses in China are family-owned, this research further considers family ownership as a moderating effect in order to examine whether this deregulation of short-selling mechanisms can work better for family businesses than other ownership identities. Lastly, Li et al. (2017) mentioned that the list of short selling deregulation only includes selected stocks rather than all stocks, which may lead to self-selection problems. Aiming to establish the reliability of empirical results, this paper will conduct propensity score matching (PSM) to compare shortable and non-shortable firms by mitigating the problem of endogeneity and self-selection and apply the difference-in-differences (DID) approach in examining the pure effect of deregulation of short selling on firm value for the listed family businesses in China.

This paper's succeeding sections begin with a literature review, which presents the theoretical foundation and hypotheses development of this research. The methodology section of this study extensively describes the data, methodology, and measures of variables.

The results and discussion parts analyze the thorough findings of the empirical study and conduct a robust test. The final part presents the conclusions, implications, and limitations of this research.

## 2. Literature Review and Hypothesis Development

While China is experiencing rapid development as the second-largest economy globally, the remarkable growth of family businesses only appeared after the transition from state-owned based economy to a socialist setting in 1992 (Cheng 2014). In this fledgeling economic environment, family businesses have severe problems concerning concentrated ownership (Yang et al. 2020), inadequate protection for minor shareholders and creditors (Gao et al. 2020), and managerial myopia for innovation investment (Xiang et al. 2019). One way to strengthen investor protection and reinforce governance mechanisms is to involve other stakeholders because of the unique Chinese business environment, in which short-sellers can supervise and punish misbehaviors of managers and family owners (Gong 2020; Massa et al. 2015).

### 2.1. Short Selling Mechanism

Two attributes are primarily expressed in determining the effects of short selling. The first aspect is that short selling acted as external governance to alleviate agency costs and promote its value (Mai and Hamid 2020). The second feature is that short selling provides pessimistic public traders with the opportunity to invest in the stock market, leading to reduced share prices and financial performance (Chen et al. 2020a). However, these two attributes of short selling are contradictory, which leads to a non-consensus about its effects on firm value creation.

Following the theory of external governance, short selling activities stimulate the detection of negative information from companies and increase the cost of seeking private benefits by managers, which can benefit from restricting their unethical behaviors and being cautious of moral hazards (Brockman et al. 2020; Chen et al. 2020b; Li et al. 2019; Mai and Hamid 2020; Rennekamp et al. 2019). For example, Chen et al. (2020b) suggested that relaxing short-selling constraints can help to reduce the risk of companies facing agency problems and information opaque, lower the borrowing costs of business loans and increase firm value, because the potential adverse effect on stock prices adds pressure on the managers of companies to act as external supervisors of the corporate governance mechanism. Brockman et al. (2020) presented empirical results from the US under Regulation SHO and stated that an increased threat of short selling could significantly improve employee relations because short-sellers may aim to short-sell stocks of firms with employee-related negative publicity. Besides, firms with a higher level of earnings manipulation, probability of labor disputes and employee whistle-blowing, and reduced workplace concerns are the targets for short-sellers. To alleviate the threats of short-sellers, companies experience better stock performance during the post Reg-SHO period.

However, only a few studies directly measure how short selling deregulation affect firm value. While short selling can act as external governance, it is still unknown whether its effect can positively improve firm value. Mai and Hamid (2020) specifically researched the effects of short selling on firm value in the Pharmaceutical Industry in China. They indicated that the appearance of short selling plays a deterrent role in corporate policies so that shortable firms engage in fewer earnings management and more productive R&D investments. In addition, Mai and Hamid (2021b) further analyzed the tourism industry and found that short-selling can improve corporate social responsibility. However, Ni and Yin (2020) criticized that companies have weaker short-run and long-run financial performance after the removal of short-sale bans. Additionally, shortable companies undertake less risk and suffer worse profitability because these companies reduce capital expenditures, cut innovation investment, and decrease external financing in the light of increasing short-selling threats.

Overall, this paper is inclined to hypothesize that short selling deregulation can improve firm value by alleviating agency cost, compliance cost, and management myopia and misconducts. Therefore, the hypotheses are examined as bellow:

**Hypothesis 1 (H1).** *There is a positive relationship between short-selling deregulation and Tobin's Q.*

**Hypothesis 2 (H2).** *There is a positive relationship between short-selling deregulation and ROA.*

Based on the viewpoint of the information detective theory, the literature demonstrated that short-sellers have superior experience and skills in collecting and analyzing public and private information compared to ordinary investors. In effect, a high volume of short selling trading offers vital signs of unfavorable risk events of the company and the predictable underperformance of the underlying stock price (Massa et al. 2015). Meng et al. (2017) further confirmed this analysis with Chinese evidence from 2010 to 2014 and stated that short-sellers have similar analyzing abilities and skills as financial analysts. Gao et al. (2018) considered the US examples and found that company insiders, such as senior managers, will take advantage of short-sellers' shorting position information to sell their shares, indicating the short-sellers are even more informative than insiders of companies. Zhou et al. (2019) reviewed the short-selling data and earning reports for listed Chinese companies from 2013 to 2016 and empirically showed that short sellers started their shorting positions just before the negative earnings report disclosure, indicating the short sellers are informed and skillful traders in detecting and predicting negative events of companies. Additionally, Chen et al. (2020b) also studied China's context before and after the 2014 short-selling ban lifting for 205 stocks and confirmed the above analysis of the prediction ability of skillful short sellers on the stock price efficiency, which implied that short-selling influences market liquidity and asymmetry of information, thereby strengthening price efficiency.

The above literature review stated that short sellers are informed investors who can detect negative information regarding companies. In contrast, corporate executives notice the accumulated short-selling positions on their stocks. Thus, they may correct their misbehaviors and announce positive hidden information to defend their stock price, which would eventually improve the stock price and firm value. Therefore, the hypotheses are postulated as bellow:

**Hypothesis 3 (H3).** *There is a positive relationship between short-selling propensity and Tobin's Q.*

**Hypothesis 4 (H4).** *There is a positive relationship between short-selling propensity and ROA.*

*2.2. Family Business*

Most of the listed private companies in China are governed by an individual shareholder or his family group (Cheng 2014; Ramos et al. 2016). However, there is no consensus about whether a family business has a premium as an ownership identity to improve the companies' firm value. The characteristics of family business can be classified into two broad categories: 'competitive advantage' and 'private benefits wedge' (Villalonga and Amit 2010). The competitive advantage hypothesis states that family ownership fosters value maximization for both family and minor shareholders (Aguiar-Inmaculada Diaz and Trujillo 2020). Under the personal benefits of the wedge hypothesis, family ownership is deeply entrenched because the family's priorities are maximized at other stakeholders' expense (Abrardi and Rondi 2020). The critical difference lies in which group of shareholders gains the highest value from the corporation. Zhu and Lu (2020) investigated the effect of family ownership on corporate environmental responsibility. Their results showed that concentrated family ownership leads to lower corporate environmental responsibility, but the negative relationship is reversed when venture capital investment comes from developed markets.

From the competitive advantage perspective, the literature has demonstrated that the Type-I agency problem, the conflict of interests between shareholders and managers, can be mitigated by close supervision on managers by family owners or even a family CEO. Tseng (2020) attempts to find the linkage of family business' long-term orientation and its SG&A expenditures by studying family firms for 11 years in the US. He hypothesized that firm value would be enhanced if SG&A expenditures are aimed for the long-term purpose of family firms. The results confirmed that, compared to non-family businesses, higher family-related ownership would positively influence innovation performance and branding management for long-lasting orientation. As a result, the Type-I agency problem is mitigated in this research, while the Type-II problem is also lessened with the alignment of interests of family members and other minor shareholders. By comparing family business, state-owned firms, and diverse-owned business in Europe from 2002 to 2011, Miroshnychenko et al. (2020) investigated how ownership structure can affect innovation inputs, combined with growth opportunities measured by Tobin's Q. They found that family ownership is more favorable for innovation investments with the moderating effect of growth opportunities than other ownership structures. Aguiar-Inmaculada Diaz and Trujillo (2020) studied Spanish family firms in the port industry. They found that concentrated family ownership can be constructive to corporate profitability measured by ROA, which is explained as confiscating behaviors practiced by the dominant shareholder in a concentrated family business because of a lack of minor shareholders. Their further research also found that the financial performance of firms in the hands of the subsequent generation of family members is more potent than that of the first-generation, indicating the diverse and professional knowledge of the next generations of family member involvement in management can contribute more to corporate profitability. The results of a study by Koji et al. (2020) showed that, when univariate analysis is used, family firms perform better than their non-family counterparts in terms of the return of assets (ROA) and Tobin's Q. Moreover, while using multivariate analysis, family firms show better performance than non-family firms with Tobin's Q. Consistent with this outcome, Ntoung et al. (2020) presented that family-owned companies have smaller financial structures than non-family-owned firms. As a result, most family-owned companies use less debt financing than non-family firms and maintain a lower level of debt.

However, from the perspective of personal benefits, family ownership enlarges the gap between family owners and minor shareholders as family owners may sacrifice the interests of the minority when necessary. Swanpitak et al. (2020b) found that family ownership in Thailand is harmful to the stock price when the society experienced unrest during the political change in 2014, which indicates that investors may discount the firm value of the family business for the bad reputation as corruption is deemed serious among family owners and politicians. As a result, the cost of high family ownership outweighed other benefits from the monitoring role of family members when the political crisis happened. By reviewing the relationship between family ownership and the financial performance of listed firms in Italy in 2000–2017, Abrardi and Rondi (2020) revealed that both concentrated ownership of family or family member as senior managers could not consummate premium on firm accounting performance. Contrary to the results of Aguiar-Inmaculada Diaz and Trujillo (2020) in Spain, the effect on stock price growth is consistent with the value-discounted theory mentioned by Swanpitak et al. (2020b). They explained that the excessive power of the controlling family and their senior family manager could destroy the firm value even with sufficient efforts on corporate governance.

The family business is also unique under the view of creditors who may offer favorable or unfriendly credit policy. By studying family businesses in Thailand, Swanpitak et al. (2020a) attempted to examine the influence of family ownership on creditors' trust by representing how family businesses can solve Type-III agency problems. The results reflected a positive relationship between the high percentage of family ownership and the cost of debt because family owners tend to achieve long-term success and invest in lower-risk projects that are preferable to creditors. Additionally, intensive ownership of

the family business can act as a supervisor for daily management involvement and enjoy higher profitability, which is also a channel for explaining the low cost of debt compared to non-family firms. Ergün and Doruk (2020) confirmed that family businesses have unique resources or attractions from creditors when acquiring external financing supports, according to Turkish empirical evidence. By studying financial constraints confronting both family business and non-family counterparts, the advantage of achieving financial sources is significantly observable for firms controlled by well-developed family groups, indicating that the family network is a vital social resource for firms to access support for growth from outside stakeholders.

For the moderating effect of family ownership, Kamaludin et al. (2020) used family ownership as moderating factors in the relationship between corporate governance and firm value of Saudi Arabian listed companies from 2012 to 2016. Surprisingly, the higher frequency and attendance of audit committee meetings are detrimental for the financial value of family firms measured by Tobin's Q, suggesting that auditing diligence in Saudi Arabia may not improve firm value. Hamid et al. (2020) demonstrated that family ownership moderated the effects of cash management and corporate governance on firm performance by studying 317 Pakistan-listed firms for 2010–2019, indicating that cash management is negatively associated with firm performance, especially among family businesses. Cordeiro et al. (2020) also examined the moderating role of the family business on the links between gender-diverse boards and firm value for stakeholders. Their findings suggest that gender-diverse boards are strongly associated with environmental value creation, in which their relationship is strengthened with the moderating effect of family ownership. Female directors display a pro-environmental stance, though family ownership leads them to focus on environmental stakeholders generally.

In China, Cheng (2014) reviewed the overall performance of family business performance and found that Chinese family owners have a substantial privilege in voting rights among the board, which can also decide the company's cash flow decision-making. With decision-making convenience, family owners are motivated to possess corporate resources at the opportunity cost of minority shareholders. Besides, the Chinese capital markets are characterized by weak investor protection and legal environment and a lack of supervision by other market stakeholders, which also induced the Type-II agency problem in Chinese family businesses. However, Tang et al. (2017) empirically stated that family ownership experienced better firm performance than non-family businesses by investigating listed companies in China from 2003 to 2014. They confirmed that family businesses could mitigate the Type-I agency problem due to lower compensation to management, lower cost of information asymmetries, and less industry influence, leading to better performance.

According to the competitive advantage theory, the presence of family ownership can alleviate agency costs and improve firm performance, while family ownership will destroy firm value creation under the perspectives of private benefits (Villalonga and Amit 2010). Nonetheless, minor shareholders will benefit from a short-selling mechanism manifested by excellent monitoring and supervision and improved investment decision and risk assessment by family owners. As most family businesses aim to retain and increase their social and economic wealth through their business, this primary objective is consistent with the competitive advantage perspectives. To achieve this objective, family owners will choose family members to lead to management positions to alleviate the agency problem and prioritize a conservative financing strategy to avoid risks and illiquidity (Harith and Samujh 2020). Based on the above reviews, this paper tends to perceive that family-owned businesses implement goal commitment aligned with the objectives of minor shareholders and management. As the agency problems in Chinese businesses widely existed and are challenging to solve due to the weak legal and economic protection for investors, the family business is a solution to balance the interests of relevant parties. As a result, a family business with better corporate governance practices in China is less threatened by the external governance of short-selling. In effect, the relationship between

short-selling and the firm value of family business is weakened. Therefore, the hypotheses are outlined as follows:

**Hypothesis 5 (H5).** *Family business weakens the relationship between the deregulation of short selling and Tobin's-Q.*

**Hypothesis 6 (H6).** *Family business weakens the relationship between the deregulation of short selling and ROA.*

**Hypothesis 7 (H7).** *Family business weakens the relationship between short-selling propensity and Tobin's-Q.*

**Hypothesis 8 (H8).** *Family business weakens the relationship between short-selling propensity and ROA.*

The hypotheses development is constructed and demonstrated in Figure 1.

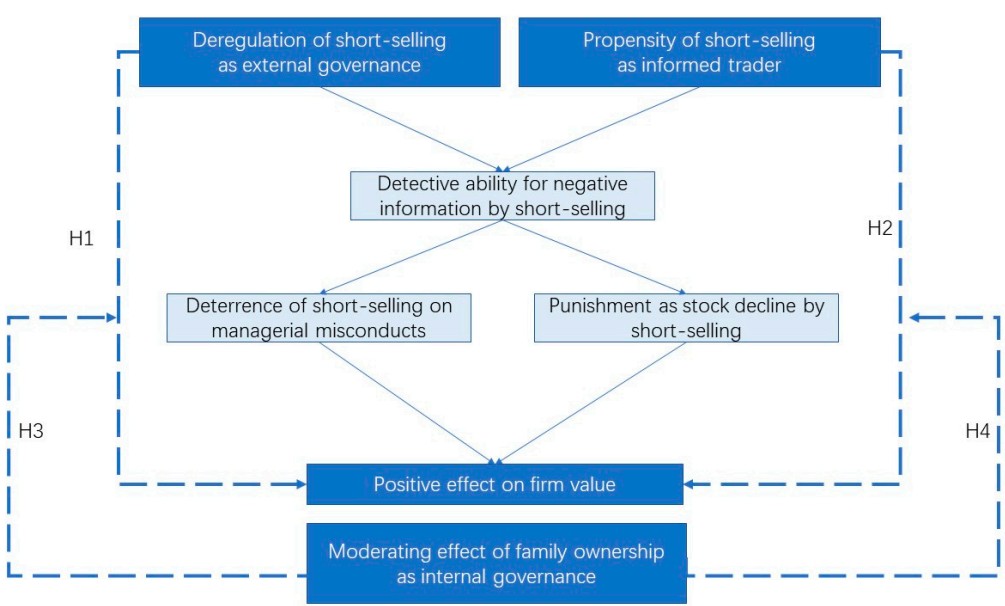

**Figure 1.** Hypotheses development.

### 3. Methodology

*3.1. Data and Sample Sources*

This paper uses data from Chinese A-share listed firms from 2009 to 2019 as the scope of the sample. The data of short selling volume and financial data are from the China Stock Market and Accounting Research Database (CSMAR). The short-sale list data is acquired from Shenzhen and Shanghai Stock Exchange websites, while the family ownership is obtained from the annual reports of listed companies. We begin the analysis in 2009 because the first round of short selling ban lifting from March 2010, so starting from 2009 ensures that no shortable stock is left in our scope of the sample.

Suggested by the common practice in developed countries (Massa et al. 2015) and China (Chen et al. 2020a), financial firms, ST* firms, firms that were previously shortable, but later became non-shortable, and firms with missing related information are excluded in this research. This paper winsorised the data at 1% and 99% levels to mitigate the effects of outliers. After the above process, we preliminarily obtained a sample of 22,468 firm-year observations, including 1405 firms in the shortable list and 1865 firms out of the shortable list.

### 3.2. Measurement of Variables and Models

This paper needs to measure both deregulation of short selling and the propensity of short sales. Following Park (2017), (Lu 2018), and Mai and Hamid (2020), SHORT * TREAT is the variable to measure deregulation of short selling. SHORT is the variable, short dummy, which is regarded as one if the stock is permitted to short during the sample period, and zero otherwise. TREAT is the time dummy that equals 1 if the stock can be shortable by the end of the current fiscal year, and zero otherwise. SHORT * TREAT is the interactive item for differences in the difference (DID) model that ensures that the firm value of a firm captures all its activities over an entire fiscal year, either before or after the exogenous shock. The interactive item SHORT * TREAT and SHORT are used as independent variables to observe the effect of short selling deregulation. Secondly, the short interest ratio (SIR) is used to measure short selling propensity, which is calculated as the number of stocks sold by short sellers minus the number of stocks repaid, divided by the trading volume of the prior day. The average SIR during a year is adopted in this paper.

For measuring firm value, this paper employed Tobin's-Q and ROA as indicators. To identify the firm value in terms of the stock market, Tobin's-Q is appropriate and is calculated as the market value of total assets divided by the asset value. While accounting for the financial value of the firm, ROA is adopted and calculated by the net income divided by the total assets.

According to previous studies (Kamaludin et al. 2020; Mai and Hamid 2020; Xiang et al. 2019), this paper adopts several sets of control variables to capture the fundamentals of companies. For financial characteristics of companies, we use firm size (SIZE) measured as a natural logarithm of the market capitalization, firm growth (GROWTH) calculated as the sales growth, leverage (LEV) measured as total liability divided by total assets, and R&D investment (RD) calculated as R&D expenditures divided by total expenses. In addition to controlling the corporate governance level of firms in the regression, we added board size (b-size) measured as the natural logarithm of the number of board members and board independence (b-ind) calculated as independent directors divided by the corporate board. To measure the identity of shareholders, we capture state-ownership (SOE) and Institutional ownership (IO), which are dummy variables (0,1). If the largest shareholder is the state or institution, then it is 1, otherwise it is 0. Year-effect (YEAR) and firm-effect (FIRM) account for time and firm-specific conditions that could play a role in the relationship between short-selling and firm value. Additionally, this paper uses robust standard errors in all models and estimations, corrected for heteroscedasticity.

For Hypotheses 1 and 2, this paper examines whether deregulation of short selling has a positive impact on firm value by using model 1 and model 2 below:

$$Tobin's\ Q = a0 + a1SHORT + a2SHORT * TREAT + a3controls + a4FIRM + a5YEAR + \varepsilon \quad (1)$$

$$ROA = a0 + a1SHORT + a2SHORT * TREAT + a3controls + a4FIRM + a5YEAR + \varepsilon \quad (2)$$

For Hypotheses 3 and 4, this paper examines whether propensity of short selling has a negative impact on firm value by using model 3 and model 4 below:

$$Tobin's\ Q = a0 + a1SIR + a2controls + a3FIRM + a4YEAR + \varepsilon \quad (3)$$

$$ROA = a0 + a1SIR + a2controls + a3FIRM + a4YEAR + \varepsilon \quad (4)$$

When measuring family ownership, this paper adopts the insights of Leitterstorf and Rau (2014) by defining family business as a type of business by which the founder and his family possess more than 25% of this total ownership. This measurement is also adopted by Zulfiqar et al. (2020) when they measured the family firms in China. Moreover, Mustafa et al. (2020) also used family ownership as a dummy variable moderator (0,1) where family firm was regarded as 1 and non-family firm is 0. In order to investigate the moderating effect of family ownership between short selling and firm value, this paper adopts an interactive item for DID model between the independent variables: deregulation

of short selling (SHORT $*$ TREAT) and short selling propensity (SIR), with the moderating variable: family ownership (FO), namely SHORT $*$ TREAT $\times$ FO and SIR $\times$ FO.

For Hypotheses 5 and 6, this paper examines whether family ownership weakens the relationship between the deregulation of short selling and firm value by using model 5 and model 6 below:

$$Tobin's\ Q = a0 + a1SHORT + a2SHORT*TREAT + a3SHORT*TREAT*FO \\ + a4controls + a5FIRM + a6YEAR + \varepsilon \tag{5}$$

$$ROA = a0 + a1SHORT + a2SHORT*TREAT + a3SHORT*TREAT*FO + \\ a4controls + a5FIRM + a6YEAR + \varepsilon \tag{6}$$

For Hypotheses 7 and 8, this paper examines whether family ownership weakens the relationship between the deregulation of short selling and firm value by using model 7 and model 8 below:

$$Tobin's\ Q = a0 + a1SIR + a2SIR*FO + a3controls + a4FIRM + a5YEAR + \varepsilon \tag{7}$$

$$ROA = a0 + a1SHORT + a2SHORT*TREAT + a3SHORT*TREAT*FO \\ + a4controls + a5FIRM + a6YEAR + \varepsilon \tag{8}$$

This paper demonstrates the definitions of the main variables in Table 1.

**Table 1.** Description and measurement of variables.

| Variables | Measurement | Sources |
|---|---|---|
| **Independent Variables** | | |
| Deregulation of short selling (SHORT) | A dummy variable (0,1), equals to 1 if the firm is shortable, and 0 otherwise | Shanghai and Shenzhen Stock exchange |
| Time factor of short selling (TREAT) | A dummy variable (0,1), equals to 1 for the years after firm is shortable, and 0 otherwise. | Shanghai and Shenzhen Stock exchange |
| SHORT*TREAT | Interactive item of SHORT and TREAT | Shanghai and Shenzhen Stock exchange |
| Short Interest (SIR) | number of stocks sold by short-sellers minus the number of stocks repaid, divided by trading volume of the prior day. | CSMAR Database |
| Family ownership (FO) | A dummy variable (0,1), equals 1 if the firm is family-owned, and 0 otherwise; Family business is defined as the founder and his family possesses more than 25% of this total ownership. | Annual Reports |
| **Dependent Variables** | | |
| Tobin's Q | Market value of total assets divided by asset value | CSMAR Database |
| ROA | Net income divided by total assets | CSMAR Database |
| **Control Variables** | | |
| Firm size (SIZE) | Natural logarithm of the market capitalisation | CSMAR Database |
| Firm growth (GROWTH) | Sales growth | CSMAR Database |
| Leverage (LEV) | Total liability divided by total assets | CSMAR Database |
| R&D investment (RD) | R&D expenditures divided by total expenses | CSMAR Database |

**Table 1.** *Cont.*

| Variables | Measurement | Sources |
|---|---|---|
| **Control Variables** | | |
| Board size (b-size) | Natural logarithm of the number of board members | CSMAR Database |
| board independence(b-ind) | independent directors divided by corporate board | CSMAR Database |
| state-ownership (SOE) | A dummy variable (0,1), equals to 1 if the firm's largest shareholder is state-owned, and 0 otherwise | CSMAR Database |
| Institutional ownership (IO) | A dummy variable (0,1), equals to 1 if the firm's largest shareholder is institution, and 0 otherwise | CSMAR Database |
| **Firm effect** | | |
| **Year effect** | | |

### 3.3. The Propensity Score Matching (PSM) Method

The shortable list of pilot stocks in China is selected based on specific standards and unique characteristics of those listed companies by the Shanghai and Shenzhen stock exchange, instead of randomly picked companies in the developed market (He and Tian 2015). To avoid the problem of self-selection, this paper attempts to use the propensity score matching (PSM) method to build an experimental group (companies in the short-sale list) and a control group (companies out of the short-sale list) that has no significant differences in the characteristics of the company in order to estimate our main specific models.

Based on the guidelines set by Caliendo and Kopeinig (2008), this paper conducted the subsequent procedures in adopting the PSM method. Firstly, we used the following stock-level characteristics of shortable firms: liquidity and volatility, firm-level characteristics (i.e., firm size, growth, leverage, and R&D expenses), and corporate governance characteristics such as board size, independence, and the identity of shareholders, as the experimental dataset and the firms out of the short-sell list as the matching sample. Secondly, we applied a logit model for the propensity score estimation in which the dependent variable is experimental. Additionally, a nearest neighbor matching strategy was conducted to achieve the closest propensity score within 0.01 to match each experimental firm with one control firm. In this method, all pairs were retained in the situation of multiple matching. Besides, the year and the industry fixed effects were considered in the model.

After the matching procedure, a total of 1405 pairs of treatment (shortable firms) and control (non-shortable firms) groups were matched with 11,234 firm-year observations. The statistical results of the differences of the after-matching treated and control firms are presented in Table 2. Before PSM, the differences among variables had been significant at the 1% level. After the process of PSM, the significance of differences dropped to 5%, 10% significance level or insignificance, with the decline of T-statistics as well. The PSM method facilitated alleviating the potentially problematic differences in company features that were known to affect corporate value and lessen concerns that the results are guided by trends in general time.

**Table 2.** Propensity score matching (PSM) results.

| | Before PSM | | | | After PSM | | | |
|---|---|---|---|---|---|---|---|---|
| **Variables** | **Treatment** | **Control** | **Diff** | **T-Statistics** | **Treatment** | **Control** | **Diff** | **T-Statistics** |
| SIZE | 22.6435 | 21.4996 | 1.1440 *** | 81.4867 | 22.6273 | 22.6304 | −0.0031 | −0.1958 |
| GROWTH | 0.1213 | 0.0433 | 0.0780 *** | 4.4002 | 0.1206 | 0.1107 | 0.0099 * | 2.9007 |
| LEV | 0.4559 | 0.3848 | 0.0711 *** | 27.5174 | 0.4555 | 0.4647 | −0.0091 | −3.4203 |
| RD | 0.0625 | 0.0913 | −0.0288 *** | −8.7765 | 0.0627 | 0.0579 | 0.0047 * | 1.6472 |
| TURNOVER | 21.4475 | 20.6484 | 0.7992 *** | 59.4930 | 21.4351 | 21.3271 | 0.1079 ** | 8.4327 |
| VOLATILITY | 6.4023 | 6.9809 | −0.5786 *** | −9.6234 | 6.3758 | 6.8646 | −0.4888 ** | −7.6344 |
| b-size | 2.1746 | 2.1098 | 0.0648 *** | 24.9848 | 2.1737 | 2.1730 | 0.0007 | 0.2598 |
| b-ind | 0.3710 | 0.3728 | −0.0018 ** | −2.4779 | 0.3710 | 0.3695 | 0.0015 * | 2.0150 |
| SOE | 0.4728 | 0.2732 | 0.1997 *** | 31.5082 | 0.4707 | 0.4851 | −0.0145 * | −2.2142 |
| IO | 0.6765 | 0.4354 | 0.2411 *** | 37.4953 | 0.6751 | 0.6410 | 0.0341 * | 5.4941 |

Note: *, **, *** Denote significance at the 10%, the 5%, and the 1%, respectively.

## 4. Results and Discussion

### 4.1. Empirical Results and Discussion

In Table 3, we describe the descriptive statistics of the key variables in this study. The means of Tobin Q and ROA are 1.7047 and 0.0483, respectively. The maximum number of Tobin Q and ROA are 6.6414 and 0.1872, while the minimum value of Tobin Q and ROA are 0.9106 and −0.1046. Due to the PSM method, the mean of SHORT is 0.5226, which indicates that about 50% of the observations are pilot firms. SIR has an average value of 0.0139, indicating 1.39% of short selling activities in the daily trading volume of observations. It is obvious that short selling trading in China is not a common tool for investors. The mean of FO is 0.3534, representing that 35% of the observations are family firms. Regarding other control variables, the mean value of firm size, growth, leverage, R&D intensity, board size, board independence, SOE and IO are 22.6289, 0.1160, 0.4601, 0.0603, 2.1734, 0.3702, 0.4779, and 0.6581 respectively.

**Table 3.** Summary statistics.

| Variables | Sample | Mean | Sd | Min | Max |
|---|---|---|---|---|---|
| Tobin's Q | 22,468 | 1.7047 | 0.8614 | 0.9106 | 6.6414 |
| ROA | 22,468 | 0.0483 | 0.0434 | −0.1046 | 0.1872 |
| SHORT | 22,468 | 0.5226 | 0.4103 | 0.0000 | 1.0000 |
| TREAT | 22,468 | 0.2523 | 0.4632 | 0.0000 | 1.0000 |
| SIR | 5680 | 0.0139 | 0.0135 | 0.0000 | 0.3442 |
| FO | 22,468 | 0.3534 | 0.4780 | 0.0000 | 1.0000 |
| SIZE | 22,468 | 22.6289 | 1.1986 | 19.7432 | 26.1355 |
| GROWTH | 22,468 | 0.1160 | 0.2750 | 0.0177 | 0.6743 |
| LEV | 22,468 | 0.4601 | 0.2035 | 0.0515 | 0.8616 |
| RD | 22,468 | 0.0603 | 0.1826 | 0.0010 | 0.9660 |
| b-size | 22,468 | 2.1734 | 0.2011 | 1.3863 | 2.8904 |
| b-ind | 22,468 | 0.3702 | 0.0535 | 0.0000 | 0.8000 |
| SOE | 22,468 | 0.4779 | 0.4995 | 0.0000 | 1.0000 |
| IO | 22,468 | 0.6581 | 0.4744 | 0.0000 | 1.0000 |

Table 4 presents the Pearson correlation matrix among variables. The short dummy (SHORT) is positively correlated with both measures of firm value, indicating that short-selling deregulation moves in the same direction of firm value, though the coefficients are only 0.2436 and 0.0641, representing a weak correlation. As the short interest ratio (SIR) is only available for shortable observation, it is not included in the correlation metrics. Family ownership is positively related to firm value, but also has a weak correlation. For the control variables, the firm growth, R&D intensity, and board independence are positively correlated with firm value. On the contrary, the firm size, leverage, board size, state-ownership, and institutional ownership are negatively correlated with market value and financial value.

**Table 4.** Pearson correlation coefficients.

| Variables | Tobin's Q | ROA | TREAT*SHORT | FO | Size | Growth | LEV | RD | b-size | b-ind | SOE | IO |
|---|---|---|---|---|---|---|---|---|---|---|---|---|
| Tobin's Q | 1.0000 | | | | | | | | | | | |
| ROA | 0.2753 *** | 1.0000 | | | | | | | | | | |
| TREAT*SHORT | 0.2436 *** | 0.0641 ** | 1.0000 | | | | | | | | | |
| FO | 0.1505 *** | 0.1923 *** | −0.0580 *** | 1.0000 | | | | | | | | |
| SIZE | −0.5104 *** | −0.1661 *** | 0.2438 *** | −0.2821 *** | 1.0000 | | | | | | | |
| GROWTH | 0.4002 *** | 0.2178 *** | −0.1311 *** | 0.1333 *** | −0.2276 *** | 1.0000 | | | | | | |
| LEV | −0.3776 *** | −0.4268 *** | 0.0275 *** | −0.2592 *** | 0.5602 *** | −0.1178 *** | 1.0000 | | | | | |
| RD | 0.0047 ** | 0.0749 *** | 0.0985 *** | 0.0376 *** | 0.0108 | −0.0542 *** | −0.0913 *** | 1.0000 | | | | |
| b-size | −0.1379 *** | −0.0349 *** | 0.0270 *** | −0.2493 *** | 0.2520 *** | −0.1455 *** | 0.1523 *** | −0.0584 *** | 1.0000 | | | |
| b-ind | 0.0182 *** | −0.0306 *** | 0.0388 *** | 0.0471 *** | 0.0320 *** | 0.0370 *** | 0.0142 ** | 0.0027 | −0.4531 *** | 1.0000 | | |
| SOE | −0.1981 *** | −0.1791 *** | 0.0480 *** | −0.7069 *** | 0.3328 *** | −0.1722 *** | 0.3170 *** | −0.0821 *** | 0.2572 *** | −0.0274 *** | 1.0000 | |
| IO | −0.0115 * | −0.1845 *** | 0.1655 *** | −0.3207 *** | 0.2156 *** | −0.2399 *** | 0.2516 *** | −0.0468 *** | 0.1238 *** | −0.0481 *** | 0.3390 *** | 1.0000 |

Note: *, **, *** Denote significance at the 10%, the 5%, and the 1%, respectively.

Table 5 illustrates the empirical findings on the relationship between short-selling deregulation, as measured by short dummy and firm value with Tobin's-Q and ROA, respectively. As hypothesised, the findings of Model 1 and 2 indicated a significant positive relationship between short-selling deregulation and ROA, as well as Tobin's Q, at the 1% level. Thus, Hypotheses 1 and 2 were supported. This result shows that short selling deregulation helps to improve the firm value of listed companies in China, indicating that, after deregulation, the appearance of short selling can act as external governance, encourage the detection of unfavourable business information, and raise the cost of demanding private benefits of executives by limiting their unethical behaviour and conflicts of interest (Mai and Hamid 2020; Massa et al. 2015).

**Table 5.** Short-selling deregulation and firm value.

| | M1a | | M1b | |
| --- | --- | --- | --- | --- |
| | Tobin's Q | | ROA | |
| **Variables** | **Coefficient** | **Probability** | **Coefficient** | **Probability** |
| Intercept | 6.8401 *** | 0.000 | −0.0966 *** | 0.000 |
| SHORT | 0.2176 *** | 0.000 | 0.0027 *** | 0.000 |
| TREAT*SHORT | 0.5087 *** | 0.000 | 0.0046 *** | 0.000 |
| SIZE | −0.2597 *** | 0.000 | 0.0081 *** | 0.000 |
| GROWTH | 11.0309 *** | 0.000 | 0.4074 *** | 0.000 |
| LEV | −0.5509 *** | 0.000 | −0.1147 *** | 0.000 |
| RD | 0.0161 ** | 0.032 | 0.0119 *** | 0.000 |
| b-size | −0.5139 *** | 0.000 | −0.0060 *** | 0.000 |
| b-ind | 0.2552 *** | 0.004 | 0.0199 *** | 0.002 |
| SOE | −0.2972 *** | 0.000 | −0.0125 *** | 0.000 |
| IO | −0.0182 * | 0.063 | −0.0135 *** | 0.000 |
| Firm-fixed | yes | | yes | |
| Year-fixed | yes | | yes | |
| R-square | 0.501365 | | 0.276921 | |
| Adj R-square | 0.500171 | | 0.272055 | |
| F-statistics | 5856.637 | | 2229.85 | |
| N | 22468 | | 22468 | |

Note: *, **, *** Denote significance at the 10%, the 5%, and the 1%, respectively. Coefficients are based on robust standard errors, corrected for heteroscedasticity.

Table 6 exemplifies the empirical findings on the relationship between short-selling regulation, as measured by short-selling propensity (SIR), and firm value. Model 3 and 4 demonstrated a positive relationship between short interest ratio and Tobin's Q, as well as ROA, significant at the 1% level, respectively. Thus, Hypotheses 3 and 4 were supported. This result indicates that the high short interest ratio may be regarded as a signal for the threats to corporate executives. To avoid further accumulating short-selling position, managers choose to correct those misconducts and eventually improve the firm value.

Table 7 demonstrates the effect of family ownership on the relationship between short-selling deregulation and firm value. In model 3a and 3b, the moderating effects were tested by adding the interaction terms (TREAT*SHORT*FO) to models 1 and 2. The results showed that the interacting terms were negatively significant at 1% level, supporting Hypothesis 5 that family ownership can act as a moderator of the relationship between short-selling and firm value. With the presence of family ownership, the firm value is dampened. This finding is consistent with Hypotheses 5 and 6. In models 4a and 4b, the moderating effects were also tested by adding the interaction terms (SIR*FO), suggesting that the interacting terms in both models were negative and significant at the 5% and 1% level. Addtionally, with respect to family business, short selling propensity does not improve firm value, which is consistent with 4a and 4b.

**Table 6.** Short-selling propensity and firm value.

| | M2a | | M2b | |
|---|---|---|---|---|
| | Tobin's Q | | ROA | |
| **Variables** | **Coefficient** | **Probability** | **Coefficient** | **Probability** |
| Intercept | 6.2755 *** | 0.000 | −0.2106 *** | 0.000 |
| SIR | 4.4419 *** | 0.000 | 0.2668 *** | 0.000 |
| SIZE | −0.1874 *** | 0.000 | 0.0133 *** | 0.000 |
| GROWTH | 22.5524 *** | 0.000 | 0.5337 *** | 0.000 |
| LEV | −1.6376 *** | 0.000 | −0.1267 *** | 0.000 |
| RD | 0.0153 * | 0.062 | 0.0122 *** | 0.000 |
| b-size | −0.4899 *** | 0.000 | −0.0062 *** | 0.000 |
| b-ind | 0.2433 *** | 0.004 | −0.0205 *** | 0.000 |
| SOE | −0.2833 *** | 0.000 | −0.0128 *** | 0.000 |
| IO | −0.0174 * | 0.060 | −0.0139 *** | 0.000 |
| Firm-fixed | yes | | yes | |
| Year-fixed | yes | | yes | |
| R-square | 0.692211 | | 0.285828 | |
| Adj R-square | 0.688559 | | 0.280504 | |
| F-statistics | 3097.965 | | 550.754 | |
| N | 5680 | | 5680 | |

Note: *, **, *** Denote significance at the 10%, the 5%, and the 1%, respectively. Coefficients are based on robust standard errors, corrected for heteroscedasticity.

**Table 7.** Short-selling, family ownership and firm value.

| | M3a | M3b | M4a | M4b |
|---|---|---|---|---|
| | Tobin's Q | ROA | Tobin's Q | ROA |
| **Variables** | **Coefficient** | **Coefficient** | **Coefficient** | **Coefficient** |
| Intercept | 7.1748 *** (0.000) | −0.1025 *** (0.000) | 5.2974 *** (0.000) | 0.2026 *** (0.000) |
| SHORT | 0.2249 *** (0.000) | 0.0027 *** (0.000) | | |
| TREAT*SHORT | 0.4073 *** (0.000) | 0.0079 *** (0.000) | | |
| TREAT*SHORT* FO | −0.0890 *** (0.000) | −0.0087 *** (0.000) | | |
| SIR | | | 4.4454 *** (0.000) | 0.3605 *** (0.000) |
| SIR*FO | | | −0.3561 * (0.051) | −0.2907 *** (0.000) |
| SIZE | −0.2377 *** (0.000) | 0.0083 *** (0.000) | −0.1675 *** (0.000) | 0.0125 *** (0.000) |
| GROWTH | 10.3988 *** (0.000) | 0.3783 *** (0.000) | 20.9496 *** (0.000) | 0.4954 *** (0.000) |
| LEV | −0.5434 *** (0.000) | −0.1062 *** (0.000) | −1.3233 *** (0.000) | −0.1298 *** (0.000) |
| RD | 0.0166 * (0.082) | 0.0121 *** (0.000) | 0.0145 * (0.071) | 0.0122 *** (0.000) |
| b-size | −0.5311 *** (0.000) | −0.0062 *** (0.000) | −0.5295 *** (0.000) | −0.0062 *** (0.000) |
| b-ind | 0.2637 *** (0.004) | −0.0204 *** (0.000) | 0.2629 *** (0.007) | −0.0205 *** (0.000) |
| SOE | −0.3071 *** (0.000) | −0.0128 *** (0.000) | −0.3062 *** (0.000) | −0.0128 *** (0.000) |
| IO | −0.0188 * (0.064) | −0.0138 *** (0.000) | −0.0188 * (0.055) | −0.0139 *** (0.000) |
| Firm-fixed | yes | yes | yes | yes |
| Year-fixed | yes | yes | yes | yes |
| R-square | 0.484504 | 0.284432 | 0.685413 | 0.27678 |
| Adj R-square | 0.492699 | 0.27627 | 0.692603 | 0.285283 |
| F-statistics | 4609.131 | 1776.662 | 2516.156 | 450.1694 |
| N | 22468 | 22468 | 5680 | 5680 |

Note: *, **, *** Denote significance at the 10%, the 5%, and the 1%, respectively. Coefficients (Probability value in parentheses) are based on robust standard errors, corrected for heteroscedasticity.

Table 8 explained the effect of short selling on firm performance by comparing family and non-family-owned companies. It is observed that deregulation of short selling is beneficial to the market values (Tobin's Q) of both family and non-family businesses, but the effect of short selling performs better for those non-family firms. However, deregulation of short selling is insignificant with the financial value (ROA) of family business, indicating that the moderating effect on short-selling constraints and ROA is not observed. Additionally, short selling propensity also significantly improves the firm value for both family and non-family businesses, but the effect is also negative on the family business firms. This finding indicates that the short-selling mechanism, which is intended to profit from detecting misconducts of business management and negative information, leads to a positive firm value by serving as external governance for all firms, including family business and non-family business. However, this external governance effect works better for non-family business. This result supports our argument that family business has better corporate governance and achieves a higher firm value (Tang et al. 2017). As a result, the appearance of short selling may not act as external governance in improving firm value for family businesses, which can alleviate the agency problem by aligning the interests of other shareholders and management. In short, when short selling is available, it is more beneficial for value creation for the non-family firms than family business.

**Table 8.** Family ownership vs. non-family ownership.

|  | FO/Non-FO | SHORT*TREAT | SIR |
|---|---|---|---|
|  |  | Coefficient | Coefficient |
| Tobin's Q | Family-owned | 0.3183 ** | 4.0893 *** |
|  | Non- Family-owned | 0.4073 *** | 4.4454 *** |
| ROA | Family-owned | −0.0008 | 0.0698 * |
|  | Non- Family-owned | 0.0079 *** | 0.3605 *** |

Note: *, **, *** Denote significance at the 10%, the 5%, and the 1%, respectively. Coefficients are based on robust standard errors, corrected for heteroscedasticity.

### 4.2. Robustness Test

#### 4.2.1. Alternative Measures of Firm Value

In this section, the researchers employ alternative measurements for the firm value as dependent variables. Specifically, PB (Price/Book value) and ROE (Return on Equity) are adopted to substitute Tobin's Q and ROA to represent market-based value and accounting-based value. Table 9 demonstrates that the interacting terms (TREAT*SHORT*FO) were negatively significant at 1% and 10% level, indicating that family ownership can act as a moderator on the relationship between short-selling and the alternative measurements of firm value (PB and ROE). However, with the presence of family ownership, the firm value decreases. In addition, the moderating effects were also tested by adding the interaction terms (SIR*FO). The results show that the interacting terms in both models were negative and significant at the 5% and 1% level, suggesting that short selling propensity does not improve alternative measurements of firm value (PB and ROE) for family business.

Table 10 explains the effect of short selling on alternative measurements of firm value (PB and ROE) with the comparison of family and non-family-owned companies. It is observed that deregulation of short selling is beneficial to the accounting value (ROE) of both family and non-family business, but the effect of short selling is better for non-family firms. However, deregulation of short selling is significant and negative in correlation with the market value (PB) of family business; thus, short selling deregulation negatively affects PB for family business. Additionally, short selling propensity (SIR) also significantly improves the alternative measurements of firm value (PB and ROE) for all firms, but the effect is also negative on family business firms. The robustness test is valid and consistent with the above analysis.

**Table 9.** Robustness test for alternative measurements of firm value.

| | PB | ROE | PB | ROE |
|---|---|---|---|---|
| **Variables** | **Coefficient** | **Coefficient** | **Coefficient** | **Coefficient** |
| Intercept | −1.0819 | −0.3905 *** | 7.8102 *** | −0.5131 *** |
| | (0.299) | (0.000) | (0.000) | (0.000) |
| SHORT | −0.4644 *** | 0.0033 *** | | |
| | (0.000) | (0.000) | | |
| TREAT*SHORT | 0.1018 * | 0.0191 *** | | |
| | (0.073) | (0.000) | | |
| TREAT*SHORT* FO | −0.1939 * | −0.0189 *** | | |
| | (0.068) | (0.000) | | |
| SIR | | | 7.5322 *** | 0.7819 *** |
| | | | (0.000) | (0.000) |
| SIR*FO | | | −2.3574 ** | −0.6833 *** |
| | | | (0.040) | (0.000) |
| SIZE | 0.0458 | 0.0203 *** | −0.2554 *** | 0.0270 *** |
| | (0.332) | (0.000) | (0.000) | (0.000) |
| GROWTH | 72.2093 *** | 0.8964 *** | 69.7196 *** | 1.0069 *** |
| | (0.000) | (0.000) | (0.000) | (0.000) |
| LEV | 1.5528 *** | −0.0638 *** | 0.6751 *** | −0.0814 *** |
| | (0.000) | (0.000) | (0.000) | (0.000) |
| RD | −0.7636 *** | 0.0129 *** | −0.7226 *** | 0.0127 *** |
| | (0.001) | (0.000) | (0.001) | (0.000) |
| b-size | 1.8901 *** | 0.0001 | 1.7885 *** | 0.0001 |
| | (0.000) | (0.781) | (0.000) | (0.769) |
| b-ind | −4.2970 *** | −0.0066 | −4.0660 *** | −0.0065 |
| | (0.000) | (0.358) | (0.000) | (0.352) |
| SOE | 0.0639 | −0.0086 *** | 0.0604 | −0.0084 *** |
| | (0.441) | (0.000) | (0.418) | (0.000) |
| IO | −2.0428 *** | −0.0106 *** | −1.9330 *** | −0.0105 *** |
| | (0.000) | (0.000) | (0.000) | (0.000) |
| Firm-fixed | yes | yes | yes | yes |
| Year-fixed | yes | yes | yes | yes |
| R-square | 0.112722 | 0.112938 | 0.8386 | 0.1434 |
| Adj R-square | 0.113555 | 0.116671 | 0.8347 | 0.1450 |
| F-statistics | 610.9079 | 605.8619 | 5416.794 | 193.0208 |
| N | 22468 | 22468 | 5680 | 5680 |

Note: *, **, *** Denote significance at the 10%, the 5%, and the 1%, respectively. Coefficients (Probability value in parentheses) are based on robust standard errors, corrected for heteroscedasticity.

**Table 10.** Family ownership vs. non-family ownership.

| | FO/Non-FO | SHORT*TREAT | SIR |
|---|---|---|---|
| | | **Coefficient** | **Coefficient** |
| PB | Family-owned | −0.0921 * | 5.1748 *** |
| | Non-Family-owned | 0.1018 * | 7.5322 *** |
| ROE | Family-owned | 0.0002 * | 0.0819 *** |
| | Non-Family-owned | 0.0191 *** | 0.0986 *** |

Note: *, **, *** Denote significance at the 10%, the 5%, and the 1%, respectively. Coefficients are based on robust standard errors, corrected for heteroscedasticity.

### 4.2.2. Multicollinearity Test Using Variance Inflation Factor (VIF)

To ensure the reliability of regression results, this section tests the multicollinearity of variance in four models of this research by adopting the variance inflation factor test (VIF). All independent variables have VIF scores that demonstrate how much this variable is explained by other independent variables. The higher the value of VIF score means the higher the multicollinearity of that independent variable. The problem of multicollinearity exists when the VIF score is higher than 10 (Hair et al. 2006). Table 11 shows that the VIF

scores of all independent variables and their mean VIF value in the four models are lower than three. Thus, the results of VIF test indicate that the problem of multicollinearity does not exist in this research.

**Table 11.** VIF scores of independent variables in four models.

|  | **Model 1** | **Model 2** | **Model 3** | **Model4** |
| --- | --- | --- | --- | --- |
| **Variables** | **VIF** | **VIF** | **VIF** | **VIF** |
| SHORT | 1.923 |  | 1.625 |  |
| TREAT*SHORT |  |  | 2.234 |  |
| TREAT*SHORT*FO |  |  | 1.558 |  |
| SIR |  | 1.153 |  | 1.558 |
| SIR*FO |  |  |  | 1.716 |
| SIZE | 2.664 | 2.162 | 2.660 | 2.175 |
| GROWTH | 1.262 | 1.316 | 1.264 | 1.310 |
| LEV | 1.560 | 1.528 | 1.562 | 1.522 |
| RD | 2.143 | 2.050 | 2.145 | 2.032 |
| b-size | 1.375 | 1.285 | 1.377 | 1.286 |
| b-ind | 1.520 | 1.412 | 1.533 | 1.425 |
| SOE | 1.367 | 1.239 | 1.552 | 1.520 |
| IO | 1.423 | 1.359 | 1.436 | 1.345 |
| Mean VIF | 1.693 | 1.500 | 1.722 | 1.589 |

## 5. Conclusions

This paper undertakes a panel data analysis with a sum of 22,468 firm-year observations from all listed companies in China from 2010 to 2019 to examine whether short selling can improve the firm value of listed companies in China. A positive relationship was found between short selling and firm value. In addition, we also tested family ownership as a moderator on the relationship between short-selling and firm value. The findings also indicated that the effect of family ownership on both short selling deregulation and short selling propensity led to an adverse firm value. Due to potential self-selection and the endogenous problem of short-selling research in China, we applied PSM-DID to measure the experimental and control groups by examining the pure effect of short selling deregulation during the policy change in firm value. The robustness test confirms the above results.

While previous literature covered research interests related to short-selling and firm value, this study extends the relevant studies in the two fields. First, the study examined these variables from the perspective of developing countries since most studies have focused on developed countries, especially in short-selling and family business research. Those empirical results from developed countries may not match the uniqueness of the Chinese capital market and business characteristics. Secondly, most studies have researched the effect of internal corporate governance, such as board independence, instead of external governance such as short selling. Thus, this paper provides a profound insight into the implication of short selling as external governance on firm value in the context of Chinese family business.

Theoretically, the findings of this paper provide new explanations on the research on the family business. Related studies stated that agency problems would persist even when internal corporate governance policies, such as audit meeting, are well conducted in the family business (Kamaludin et al. 2020). Based on the insights of Villalonga and Amit (2010), competitive advantage theory explains that family owners closely monitor and supervise corporate executives, which can mitigate the Type-I agency problem between shareholders and managers. Moreover, the private benefits theory argued that family owners would overlook the interests of minor shareholders with their advantageous power, which aggravates the Type-II agency problem between major and minor shareholders. In this research, the findings demonstrate that the benefits of family ownership play a significant role in corporate management by aligning the interests of other shareholders. Despite the advantages of family businesses, the availability of short selling is not a useful

mechanism in acting as external governance and in improving the firm value of family businesses in the Chinese context.

This research has several practical implications. Firstly, from the perspective of Chinese policymakers, the benefits of short selling deregulation as external governance on firm value encourage the Chinese government to expand the scale of short-selling ban and lift regulations, especially for non-family businesses. Secondly, from the perspective of financial regulators in developing countries, the empirical results of this study show the positive effects of short selling, so those countries with weak corporate governance and inadequate regulation protection similar to China can try to implement a pilot program of short selling deregulation, which is perceived to complement corporate governance mechanisms in China. Thirdly, from the perspective of family businesses, the competitive advantage of family control is more advanced than non-family businesses in terms of corporate governance, so short-selling deregulation may not be a real threat to family businesses. However, family businesses still need to explore the potentials of external governance mechanisms as independent monitoring mechanisms on the behaviors of family members though short selling is not the appropriate mechanism for family business.

The findings of this study have certain limitations with regard to the context and methodology. Firstly, the empirical results only reflect the relationship of short-selling, family business, and firm value of listed family companies in China. Likewise, the percentage of family ownership is not examined in this research. Besides, this analysis has been carried out in China as an example of a developing country. Thus, it is also essential to compare the effects of short selling between developed and developing countries. Finally, the firm value in this research is only proxied by the financial value and market value, but other values for stakeholders, such as innovation performance and environment, society, and governance (ESG) performance, are not included. These are essential for policymakers to consider in further lessening short-selling constraints if short selling can also improve non-financial value for stakeholders. Based on the limitations of this study, future research can focus on the following directions. Firstly, the further analysis shall consider the percentage of family ownership as a moderating variable since a high or low percentage of family ownership may have a different moderating effect on short selling and firm value. Secondly, further investigations can be carried out by comparing empirical results between developed and developing countries such as Hong Kong and Mainland China, which are also available for short selling and have similar cultures and business characteristics, but have different capital market development levels. Thirdly, future studies can adopt non-financial factors, such as ESG or innovation performance, to measure the firm value of listed companies.

**Author Contributions:** Writing—original draft, W.M.; Writing—review & editing, N.I.N.B.A.H. All authors have read and agreed to the published version of the manuscript.

**Funding:** This research received no external funding.

**Institutional Review Board Statement:** Not applicable.

**Informed Consent Statement:** Not applicable.

**Data Availability Statement:** The data of this study is available from the authors upon request.

**Conflicts of Interest:** The authors declare no conflict of interest.

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
