# Peer review of "The Moderating Effect of Family Business Ownership on the Relationship between Short-Selling Mechanism and Firm Value for Listed Companies in China"

_jrfm, doi:10.3390/jrfm14060236_

Round 1

Reviewer 1 Report

The paper is well written, the bibliography used is much updated, the research design and methodology are appropriated and the results are clearly presented. However, I suggest some issues to improve the paper.

  • When authors use family ownership as a moderating variable, it seems that they are going to analyse how family businesses act if they have more or less family ownership, that is, the percentage of ownership in family firms. However, what the authors are studding is the difference between family and non-family business, as the dummy variable is 1 (family business) and 0 (non-family business) and not in function of family ownership. Please, try to clarify this issue because the authors are actually analysing family business vs non-family businesses.
  • Related to the previous paragraph, to complete this paper, or maybe in a future paper, the authors could analyse the percentage of family ownership as a moderating variable among the family businesses. Specially, to study if family businesses with a higher family ownership weaken more the relationship between short-selling and firm value and if family businesses with a lower family ownership weaken lees such a relationship. It is a way to study the heterogeneity of family businesses, as not all the family businesses are the same.
  • Figure 1 is not clear, draw in a more visual and clear way.
  • Practical implications are scarce or missing.
  • As this analysis has been carried out in developed countries, it would be interested to make a comparison between developed and developing countries.
  • Finally, I recommend to city papers from this journal (Journal of Risk and Financial Management)

Reviewer 2 Report

Please see my comments in the file attached. 

Reviewer 3 Report

The focus of the paper is explicitly described and, in the current version, it is quite clear the positioning of the manuscript, in the present academic debate.

From the methodological standpoint, it is worthwhile giving more examination to the reliability of the empirical evidence (Sections 3). In particular, I heartly prompt to show the results pertinent to the robustness tests (i.e. Hausman, VIFs and so on). Still, did the Authors compute robust standard errors?

With reference to the structure, the paper seems well balanced. Nonetheless, it is necessary to conduct an accurate proofreading, in order to detect some typos, such as:

  • “Based on viewpoints of information detective theory. literatures”, see line 166;
  • “??? = ?0 + ?1???? ∗ ????? + ?2????? ∗ ????? ∗ ?? +”, see line 410;
  • “demonstrated a postive relationship”, see line 503;
  • “short selling constraints and ROA is not oberseved”, see line 534;
  • “owners are threats for effective corporate governancen and”, see line 547;
  • “family business and firm value. literatures stated”, see line 582.

At the current stage, in my opinion, this manuscript is able to give a prominent support for moving the extant body of knowledge forward. Indeed, good foundations and huge potentialities emerge with respect to the degree of novelty perceivable by your work.

Therefore, I recommend you to carry out a major revision of your manuscript with the aim to improve the current version. Good luck!

Round 2

Reviewer 2 Report

Thank you for responding to my comments sufficiently. Congratulations for this nice work.

Author Response

Thank you very much for your comments. 

Reviewer 3 Report

After having read the updated version of the manuscript, the weakness pertinent to the reliability of the empirical evidence however persists.

In more detail, the Authors did not apply any robusteness test suggested in my prior revision, such as Hausman, VIFs and so on. They just included the Section 4.2 (see p. 16) where there is a mere change of the dependent variables rather than the calculation of the foregoing tests. Therefore, in my opinion, once again the paper should be revised, in order to gain the final goal of the publication.

Good luck!

Round 3

Reviewer 3 Report

After having read, for the second time, the new version of the manuscript, the weakness inherent to the reliability of the empirical evidence persists. Very briefly, the Authors just applied the VIF tests and included a specific Section (i.e. 4.2.2).

Unfortunately, there is still no mention with respect to Hausman test. In this regard, once again, I emphasise the relevance of the foregoing robustness test, as the Authors carried out a longitudinal analysis and it is necessary to corroborate the methodological choice pertinent to the fixed effects.

That said, I recommend the Authors to follow or better to consider all the comments provided by the reviewers. In the case they disagreed, it is reasonable to give an answer and, at the same time, to corroborate their point of view. In so doing, there is a mutual enrichment from the review of a manuscript.

Therefore, for the last time, I repeat my opinion for which the paper should be revised, in order to gain the final goal of the publication.

Good luck!
